# Antibacterial Thiopeptide GE2270-Congeners from *Nonomuraea jiangxiensis*

**DOI:** 10.3390/molecules28010101

**Published:** 2022-12-23

**Authors:** Kuan-Chieh Ching, Elaine J. Chin, Mario Wibowo, Zann Y. Tan, Lay-Kien Yang, Deborah C. Seow, Chung-Yan Leong, Veronica W. Ng, Siew-Bee Ng, Yoganathan Kanagasundaram

**Affiliations:** Singapore Institute of Food and Biotechnology Innovation (SIFBI), Agency for Science, Technology and Research (A*STAR), Singapore 138673, Singapore

**Keywords:** thiopeptide, GE2270, antibacterial, natural products, RiPPs, structure elucidation

## Abstract

Thiopeptides are macrocyclic natural products with potent bioactivity. Nine new natural thiopeptides (**1**–**9**) were obtained from a *Nonomuraea jiangxiensis* isolated from a terrestrial soil sample collected in Singapore. Even though some of these compounds were previously synthesized or isolated from engineered strains, herein we report the unprecedented isolation of these thiopeptides from a native *Nonomuraea jiangxiensis*. A comparison with the literature and a detailed analysis of the NMR and HRMS of compounds **1**–**9** was conducted to assign their chemical structures. The structures of all new compounds were highly related to the thiopeptide antibiotics GE2270, with variations in the substituents on the thiazole and amino acid moieties. Thiopeptides **1**–**9** exhibited a potent antimicrobial activity against the Gram-positive bacteria, *Staphylococcus aureus* with MIC_90_ values ranging from 2 µM to 11 µM. In addition, all compounds were investigated for their cytotoxicity against the human cancer cell line A549, none of the compounds were cytotoxic.

## 1. Introduction

Thiopeptides (or thiazolyl peptides) are a class of ribosomally synthesized peptides that share a common pyridine/piperidine ring decorated by numerous azoles and dehydroamino acids [1]. These complex secondary metabolites are known to exhibit a potent inhibition against Gram-positive bacteria, with over 100 structures reported thus far [1,2]. The expanded understanding of thiopeptide biosynthesis and their mode of actions, as well as the emergence of bacterial resistance, has led to an increased interest in this class of antibiotics [2,3], with several compounds successfully entering the market, such as thiostrepton [4] and nosiheptide [5] (Figure 1), for veterinary applications. Furthermore, LFF571, a semi-synthetic analog of GE2270A successfully reached a phase 2 clinical trial for treating *Clostridium difficile* infections in humans, but the trial was discontinued in 2019 because LFF571 showed higher recurrence rates than conventional antibiotics treatment with vancomycin [6,7]. Nevertheless, these compounds sufficiently exemplify the potency of thiopeptides as promising candidates for antimicrobial drug discovery. Thus, it would be of interest to investigate new thiopeptide analogs in an effort to discover new lead compounds. 

As part of our on-going studies in biologically active secondary metabolites from Actinobacteria [8,9,10], several thousand extracts from our in-house library [11] were screened for their antibacterial activity, with a few extracts demonstrating antibacterial activity against *Staphylococcus aureus*. The HR-ESIMS and Global Natural Products Social (GNPS) molecular networking analysis of these extracts revealed the presence of several potentially unknown thiopeptides structurally related to GE2270. Further chemical analysis of the active extracts shortlisted one extract from the *Nonomuraea jiangxiensis* strain A7611 that produced higher yields of potentially new thiopeptide analogs for large-scale cultivation for bioactive compound identification. In this work, the isolation and structure determination of nine new naturally occurring thiopeptides (**1**–**9**) from *Nonomuraea jiangxiensis*, as well as their antibacterial activity were evaluated and reported. The discovery of these new compounds not only enriched the structural diversity of thiopeptide antibiotics, but also provided insights into preliminary structure–activity relationships (SAR). 

## 2. Results and Discussion

The HPLC-MS analysis of the extract of *Nonomuraea jiangxiensis* strain A7611 found several masses of sulfur and nitrogen containing compounds, indicating the presence of thiopeptide. To visualize the overall chemical space in the extract, a HPLC-MS/MS experiment was done. The MS/MS data were used to generate a consolidated GNPS molecular network [12]. In this molecular network, specific clusters containing potentially new thiopeptides were found (Figure 2). To identify the thiopeptides, a large-scale fermentation was performed, and the CH_2_Cl_2_ extract was subjected to RP-HPLC and PTLC to obtain compounds **1**–**9** (Figure 3). 

Compound **1** (Figure 2) was isolated as a white amorphous powder and its molecular formula was established as C_56_H_54_N_14_O_11_S_6_ based on HR-ESIMS measurements. The ^1^H NMR data (Table 1) revealed features of a peptide-derived compound, including five amide ^1^H signals (δ_H_ 9.29, 8.69, 8.69, 8.45, 7.41). The ^13^C NMR data (Table 1) were also consistent with a peptide-derived compound, comprising of oxazoline and thiazole units including five amide carbonyls and one carboxylic acid carbonyl signals (δ_C_ 169.4, 169.3, 163.2, 161.2, 161.0, 160.2), six thiazole (δ_C_ 170.8, 168.3, 167.9, 165.4, 164.5, 160.3) and oxazoline (δ_C_ 160.1) moieties. The presence of thiazole rings A, B, C, D, E and F, pyridine (Py), phenylserine (PheSer), valine (Val), glycine (Gly), asparagine (Asn), oxazoline (Oxa) and proline (Pro) residues in **1** was supported by the observed COSY and HMBC correlations (Figure 4). A further analysis of the 2D-NMR data revealed that the structure of **1** was similar to that of GE2270A, a thiopeptide isolated from *Planobispora rosea*. The NMR data comparison between **1** and those of GE2270A showed a high degree of similarity, implying their structural analogy [13,14]. The 1D NMR resonances of the aromatic protons of Py and of the thiazole rings A, B, C, D, E and F for GE2270A and **1** were coincided [13,14]. This showed that the building blocks of the thiazolyl peptide backbone and the sequence was conserved. Even though the structure of **1** was previously reported from a biosynthetic gene cluster and heterologous expression studies, the NMR spectroscopic data of **1** were not reported [15]. A detailed inspection of the ^13^C NMR data of **1** (Table 1) together with a comparison of the reported NMR data of GE2270A revealed a missing signal of δ_C_ 173.5 that corresponded to the C-terminal amide/carbonyl of proline (Pro) residue in GE2270A, which was replaced by a carboxylic acid group (δ_C_ 169.3) in **1** [13,14]. Based on these data, the structure of **1** was established to be an analog of GE2270A possessing a C-terminal carboxylic acid. Of note, this was the first report on the isolation of **1** from a native strain.

Compound **2** (Figure 3) was isolated as white amorphous powders. The HR-ESIMS measurements determined the molecular formula of **2** as C_54_H_50_N_14_O_10_S_6_. Furthermore, compounds **3** and **4** (Figure 3) were also isolated as white amorphous powders and their molecular formulae were established as C_53_H_48_N_14_O_10_S_6_ and C_55_H_52_N_14_O_11_S_6_, respectively. ^1^H NMR data of **2**–**4** (Table 1 and Appendix A) were consistent with that of **1** with minor differences. For instance, the methyl singlet (δ_H_ 2.59) on thiazole ring E in **1** was not observed in the ^1^H NMR spectra of **3** and **4**. This implied that C-5 of thiazole ring E in **3** and **4** was unsubstituted, which was further supported by an additional singlet at around δ_H_ 8.10 in the ^1^H NMR spectra of **3** and **4**. On the other hand, the singlets at δ_H_ 3.39 and 4.99 of the methoxymethyl substituent at the C-5 position of thiazole ring D were not detected in **2** and **3**. This observation suggested that thiazole ring D was unsubstituted at the C-5 position supported by an additional singlet at δ_H_ 8.29 in the ^1^H NMR spectra of **2** and **3**. The remaining structural moieties in compounds **2**–**4** were established by COSY and HMBC correlations (Figure 4 and Appendix A). Compounds **1**–**4** shared the same thiazolyl peptide backbone and C-terminal carboxylic acid group while possessing altered decorations at thiazole rings D and E, they are grouped together and named GE2270F_1_, GE2270F_2_, GE2270F_3_ and GE2270F_4_, respectively. 

Compounds **5** and **6** (Figure 3) were isolated as white amorphous powders and their molecular formulae were established as C_54_H_54_N_15_O_10_S_6_ and C_56_H_58_N_15_O_11_S_6_, respectively, based on HR-ESIMS measurements. Interpretation of the COSY, HSQC and HMBC spectra of **5** and **6** together with the NMR data comparison with those of GE2270A revealed the replacement of Oxa group with a serine (Ser) moiety, which was indicated by an amide carbonyl at δ_C_ 173.5 (Appendix A) [13,14]. The presence of a Ser residue was evident from the COSY correlations between NH (δ_H_ 8.47)/H-α (δ_H_ 4.90), H-α (δ_H_ 4.90)/H_2_-β (δ_H_ 3.76, 3.82), and H_2_-β (δ_H_ 3.76, 3.82)/OH (δ_H_ 5.28), as well as the HMBC correlation from H-α to a carbonyl carbon at δ_C_ 168.9 (Appendix A). Compounds **5** and **6** were likely to be intermediates or precursors in the biosynthesis of GE2270C_1_ and GE2270A, respectively, as reported in the literature through the cyclization of Ser into an Oxa group catalyzed by a YcaO-like cyclodehydratase enzyme [15]. Compound **6** was previously synthesized and tested for antibacterial activity in the search of a lead thiazole peptide that has an enhanced aqueous solubility, and the activity against *S. aureus* was equivalent to that of GE2270A but with a poor aqueous solubility [16].

Compounds **7**, **8** and **9** (Figure 3) were isolated as white amorphous powders and their molecular formulae were determined as C_47_H_42_N_12_O_8_S_6_, C_49_H_46_N_12_O_9_S_6_ and C_48_H_44_N_12_O_9_S_6_, respectively. Compounds **7**–**9** possessed the same thiazolyl peptide backbone as in **1** with the loss of Pro and Oxa groups as indicated by the missing proton and carbon signals that correspond to Pro and Oxa groups (Figure 3 and Appendix A). Thiopeptides **7** and **8** were analogs of one another with or without the substitution at the C-5 position of thiazole ring D, and both possessed the terminal methyl–ester functionality, which is indicated by an additional singlet at δ_H_ 3.91 (Appendix A). On the other hand, **9** was the free carboxylic acid form of **8**. These compounds were likely to be intermediates or precursors in the biosynthesis of GE2270A. Compounds **8** and **9** were previously reported and synthesized as intermediates in the total synthesis of thiopeptide GE2270 analogs [17,18]. This was the first report of the isolation of thiopeptides **8** and **9** from Nature. Notably, one close structural analog of compounds **8** and **9**, which possessed a terminal amide group in place of either an ester or a carboxylic acid group was previously synthesized and tested for antibacterial activity and was found to be inactive against Gram-positive *S. aureus* [17]. 

Although the thiopeptide GNPS molecular clusters showed several related thiopeptides, it was not possible to isolate and acquire the NMR spectroscopic data for two minor compounds (*m*/*z* 1081.15 for **10**, and *m*/*z* 1219.19 for **11**) due to their minute amounts in the extract. Nevertheless, their molecular formulae were determined by exact mass calculation (Table 2). Due to the small amount of the samples, we were not able to assign an absolute configuration using Marfey’s reagent. However, the structures of thiopeptides **1**–**9** are closely related to a series of known thiopeptides, GE2270. Further manual observation of the HPLC-HRESIMS/MS of the crude extract indeed revealed the presence of GE2270A as shown by a molecular ion *m*/*z* 1290.2663 [M+H]^+^ consistent with that of GE2270A (Appendix A) [15]. Thus, **1**–**9** were most likely to be biosynthetically related to GE2270A, a thiopeptide initially isolated from *Planobispora rosea* [19]. Although the structures of compounds **1**–**9** and GE2270A contain several typical characteristics of a nonribosomal peptide (NRP), GE2270A is a ribosomally synthesized, post-translationally modified peptide (RiPP), which was also observed in other known thiopeptides [15]. Therefore, **1**–**9** was presumed to occur in the configuration, as shown in Figure 3, and determined to be new members of GE2270 thiopeptides [20]. This was further supported by a comparison of the specific rotation and the ^1^H and ^13^C NMR data of **1**–**9** with those of GE2270A [19,21], whose configuration was confirmed by total synthesis [18,22].

Compounds **1**–**9** were evaluated for their antibacterial activity against a panel of bacterial strains, namely *A. baumannii* (ATCC^®^ 19606™), *K. aerogenes* (ATCC^®^ 13048™), *P. aeruginosa* (ATCC^®^ 9027™) and *S. aureus* Rosenbach (ATCC^®^ 25923™). Thiopeptides **1**–**9** were found to be inactive against Gram-negative bacterial strains (Appendix A), while **1**, **2**, **6**, **7**, **8** and **9** displayed activities against *S. aureus* Rosenbach (ATCC^®^ 25923™), the only Gram-positive bacterial strain tested (Table 3 and Figure 5). Interestingly, **3**, **4** and **5** were found to be inactive against *S. aureus*. In addition, the antifungal effects of thiopeptides **1**–**9** were also determined against *Aspergillus fumigatus* (ATCC^®^ 46645™), and no antifungal activity was observed (Appendix A). Furthermore, all the compounds were tested for their cytotoxicity against the human lung carcinoma cell line A549 (ATCC^®^ CCL-185™); none of the compounds were cytotoxic towards A549 cells (Appendix A). Notably, it was well documented that thiopeptides exhibited a wide range of biological properties and are strong antibiotics against Gram-positive bacteria [1], including contemporary strains of methicillin-resistant *Staphylococcus aureus* (MRSA).

Compounds **1**–**4** possessed the same R1 group on thiazole ring A, but various substitutions, i.e., the R2 and R3 groups on thiazole rings D and E, respectively. As shown in Table 3, **1** was three-fold more active (i.e., minimal inhibitory concentration (MIC_90_) of 2.63 µM and minimal bactericidal concentration (MBC_90_) of 18.48 µM) against *S. aureus* Rosenbach (ATCC^®^ 25923™) than **2** (i.e., MIC_90_ of 6.94 µM and MBC_90_ of 67.93 µM). This was likely attributed to the presence of both the R2 (i.e., methylene-oxy-methyl) and the R3 (i.e., methyl) groups in **1**, resulting in a significant increase in antibacterial activity. In addition, the presence of the R3 group and the absence of the R2 group in **2** resulted in a weaker antibacterial activity. The absence of both R2 and R3 groups in **3**, and the absence of the R3 group in **4** resulted in no bioactivity. Compounds **5** and **6** shared similar chemical structures, each possessing a terminal serine–proline group (R1) on thiazole ring A. Compound **6** possessing both R2 and R3 groups demonstrated antibacterial activity against *S. aureus* Rosenbach (ATCC^®^ 25923™), while **5** possessing only the R3 group displayed no activity against *S. aureus* Rosenbach (ATCC^®^ 25923™). This indicated that the presence of both R2 and R3 groups was important for the bioactivity of this series of thiopeptides.

Both compounds **7** and **8** possessed a terminal ester group (R1) on thiazole ring A, with **7** possessing only the R3 group, whilst **8** possessed both R2 and R3 groups. A slight increase in the antibacterial activity (i.e., MIC_90_ of 3.17 µM) was observed in **8** when compared to **7** (i.e., MIC_90_ of 4.71 µM) when both R2 and R3 groups were present. Compound **9** is the only compound with a terminal carboxylic acid group (R1) on thiazole ring A. Interestingly, at least a three-fold reduction in the antibacterial activity (i.e., MIC_90_ of 10.27 µM) was observed in **9** when compared to **8** when the ester group was changed to a carboxylic acid. This could be due to the poor cell membrane permeability of the carboxylic acid group, thus resulting in a poorer antibacterial activity. The mechanism of action of thiopeptides was previously studied, and it was well established that thiopeptides exert their antibacterial function in the bacterial cell via the inhibition of ribosomal protein synthesis [2].

A nucleotide BLAST search of the 16S rRNA gene sequence of A7611 was performed against the NCBI 16S ribosomal RNA database revealed that the isolate shared 99.93% sequence identity (E-value = 0.0) to 16S rRNA of *Nonomuraea jiangxiensis* with accession number NR_116645.1. The phylogenetic relatedness using the neighbour-joining analysis method of the isolated strain and its closely related species was obtained from the GenBank database and is shown in Figure 6.

## 3. Materials and Methods

### 3.1. General Experimental Procedures

A JASCO P-2000 digital polarimeter was utilized to measure the specific rotations of the compounds. A Bruker DRX-400 NMR spectrometer was utilized to obtain the NMR spectra of the compounds. Specifications of the NMR spectrometer include a Cryoprobe, and a 5 mm BBI (1H, G-COSY, multiplicity-edited G-HSQC, and G-HMBC spectra) or BBO (13C spectra) probe heads equipped with z-gradients. Residual solvent peaks for DMSO-*d_6_* were set at δ_H_ 2.50 and δ_C_ 39.5 ppm as reference signals in the ^1^H and ^13^C NMR spectra, respectively. A preparative HPLC experiment was performed using Agilent 1260 Infinity Preparative-scale LC/MS Purification System coupled to an Agilent 6130B single quadrupole mass spectrometer with an XTerra Prep MS C_18_ column (19 × 300 mm, 10 µm). The detection wavelength used in the preparative HPLC was 254 nm. An Agilent UHPLC 1290 Infinity, coupled with an Agilent 6540 accurate–mass quadrupole time-of-flight (QTOF) mass spectrometer, equipped with an ESI source was utilized to conduct the HPLC-MS experiment. The analyses were conducted with an Acquity UPLC BEH C18 column (2.1 × 50 mm, 1.7 µm), at a flow rate of 0.5 mL/min under standard gradient conditions of 2% MeCN (0.1% formic acid) to 100% MeCN (0.1% formic acid) over 8.6 min. The operating parameters for QTOF were the same as previously reported [8]. 

### 3.2. Molecular Identification and Phylogenetic Analysis of the Bacteria Isolate A7611

The bacterial strain A7611 was isolated from terrestrial soil in Singapore. The isolated bacteria was grown on Bennett Agar for 5–7 days at 28 °C. The DNA of the strain was extracted from the plate using the DNeasy PowerSoil Pro Kit (Qiagen, Hilden, Germany) according to the manufacturer’s protocol where the cells underwent a beat-beating step for cell disruption using an automated tissue homogenizer and cell lyser 1600 MiniG (SPEX SamplePrep, Metuchen, NJ, USa) at 1500 rpm for 3 min. The NanoDrop2000 spectroscopy system (ThermoFisher Scientific, Waltham, MA, USA) was used to measure the DNA purity and yield extracted. Bacterial 16S rRNA genes were amplified from the DNA extracted from the isolated actinobacteria with universal 16S primers 27F (5′-AGA GTT TGA TCC TGG CTC AG-3′) and 1492R (5′-TAC GGY TAC CTT GTT ACG ACT T-3′) [23,24]. The PCR amplification reactions were performed using Applied Biosystems ProFlex Thermocycler (ThermoFisher Scientific, Waltham, MA, USA) with a total reaction of 20 µL that comprised of 2.0 µL of 10× PCR buffer with 20 mM MgCl_2_, 2.0 µL of 2 mM dNTPs, 1 unit of Taq polymerase (ThermoFisher Scientific, Waltham, MA, USA), 1.0 µL of 10 µM of each primer and 1.0 μL of purified DNA templates. A non-template and a negative control using sterile resuspension buffer were included in the run. The reactions were subjected to the following temperature cycling profile of initial denaturation at 95 °C for 5 min; 30 cycles each of 30 s at 95 °C for denaturation; 50 s at 60 °C for annealing and 1 min at 72 °C for extension, with a final extension of 5 min at 72 °C. The commercial service of forward and reverse Sanger Sequencing was performed on the PCR amplified DNA fragment (1st BASE, Singapore, Singapore). The sequences obtained were aligned using Benchling and further analyzed using BLAST (National Center for Biotechnology Information (NCBI)). Related actinobacteria strains, retrieved from the GenBank databases, were aligned with the sequence of the I6S rRNA region of the isolated strain A7611 using ClustalW. A neighbor-joining tree algorithm method was used to determine the genetic relationship between the strains. The phylogenetic tree was constructed with a bootstrapped database containing 1000 replicates in MEGA 11.0 software (Mega, State College, USA). The DNA sequence for the sample A7611, reported in the present study, were deposited with GenBank database of NCBI under the accession numbers OM967343.

### 3.3. Fermentation and Extraction of Bacterial Crude Extract

*Nonomuraea jiangxiensis* strain A7611 was cultured in 5 mL SV2 media, (For 1 L, add 15 g glucose (1st BASE, Singapore, Singapore), 15 g glycerol (VWR, Radnor, PA, USA), 15 g soya peptone (Oxoid, Basingstoke, Hampshire, UK), and 1 g calcium carbonate (Sigma-Aldrich, St. Louis, MO, USA), pH was adjusted to 7.0) for 3 days at 28 °C with shaking performed at 200 rpm. Saturated seed cultures were diluted in fresh fermentation media: CA09LB (For 1 L, add 10 g meat extract (Sigma-Aldrich, St. Louis, MO, USA), 4 g yeast extract (BD Biosciences, Franklin Lakes, NJ, USA), 20 g glucose (1st BASE, Singapore, Singapore), and glycerol 3 g (VWR, Radnor, PA, USA), pH was adjusted to 7.0) in a 1:20 volume ratio and fermented with 200 rpm shook at 28 °C in the dark. The cultures were pelleted after 9 days followed by lyophilization of the separated biomass and supernatant. The dried samples were extracted by MeOH then filtered through filter paper (Whatman Grade 4, Maidstone, Kent, UK). MeOH was removed under reduced pressure to give a crude extract of a combined weight of 15.70 g. The crude extract consists of broth extract 84.4% (13.25 g) and biomass extract 15.6% (2.45 g).

### 3.4. Isolation and Structure Elucidation

The dried extracts obtained were combined and partitioned with CH_2_Cl_2_/MeOH/H_2_O in a ratio of 2:1:1. A rotary evaporator was utilized to remove the CH_2_Cl_2_ layer under a reduced pressure. The CH_2_Cl_2_ crude extract (692 mg) was redissolved in CH_2_Cl_2_ and subjected to a silica gel column chromatography (Merck, Silica gel 60, 0.040–0.063 mm). The column was eluted with a stepwise gradient of 0%, 2%, 4%, 8%, 10% and 12% MeOH in CH_2_Cl_2_ followed by 100% MeOH. The 12% MeOH in CH_2_Cl_2_ and 100% MeOH fractions were combined to obtain 350 mg of an enriched fraction of thiopeptide analogs. The dried extract was then redissolved in CH_2_Cl_2_:MeOH in a ratio of 1:1. Further separation was performed using a Sephadex LH-20 column (mobile phase: CH_2_Cl_2_:MeOH = 1:1) to obtain subfraction, containing thiopeptide analogs (53 mg). The dried mixtures were dissolved in MeOH and separated by C_18_ RP-HPLC (solvent A: H_2_O + 0.1% HCOOH, solvent B: acetonitrile + 0.1% HCOOH; flow rate: 24 mL/min, gradient conditions: 70:30 isocratic for 5 min; 30% to 60% of solvent B over 55 min, 60% to 100% of solvent B over 2 min, and finally isocratic at 100% of solvent B for 10 min to give 5.1 mg of compound **2** (RT = 30.5 min), 6.9 mg of compound **1** (RT = 37 min), 1.1 mg of compound **7** (RT = 45 min), 0.2 mg compound **8** (RT = 51 min), and 0.5 mg of compound **9** (RT = 39 min). Fractions collected between retention the times of 23.5 min and 27 min were combined. The dried combined fraction was separated by preparative normal phase TLC (Merck, TLC Silica gel 60 F_254_, 20 × 20 cm, 6% MeOH/CH_2_Cl_2_ for the first TLC run followed by 10% MeOH/CH_2_Cl_2_ for the second elution) to obtain **5** (2.0 mg) and **3** (1.0 mg). In addition, fractions collected between retention times of 31.5 min and 33.75 min were combined. The dried combined fraction was further separated by preparatory TLC using the same conditions used for the separation of **5** and **3**, to obtain **6** (2.7 mg) and **4** (1.7 mg). 

### 3.5. Chemical Structural Data

The UV spectra, HRESIMS spectra, 1D and 2D NMR spectra of **1**–**9** are provided in Appendix A.

GE2270F_1_ (**1**): White amorphous powders; ${\left[\alpha \right]}_{D}^{23}$ + 78 (c 1.2, CH_2_Cl_2_:MeOH = 1:1); UV (MeCN/H_2_O) λmax (%) 221 (100%), 308 (33%) nm; 
(+)-HRESIMS: *m*/*z* 1291.2544 [M+H]+ (calcd for C_56_H_55_N_14_O_11_S_6_, 
1291.2499); ^1^H and ^13^C NMR data, see Table 1.

GE2270F_2_ (**2**): White amorphous powders; ${\left[\alpha \right]}_{D}^{23}$ + 111 (c 0.9, CH_2_Cl_2_:MeOH 
= 1:1); UV (MeCN/H_2_O) λmax (%) 221 (100%), 308 (33%) nm; 
(+)-HRESIMS: *m*/*z* 1247.2240 [M+H]+ (calcd for C_54_H_51_N_14_O_10_S_6_, 
1247.2237); ^1^H and ^13^C NMR data, see Table 1.

GE2270F_3_ (**3**): White amorphous powders; ${\left[\alpha \right]}_{D}^{23}$ + 58 (c 0.2, CH_2_Cl_2_:MeOH 
= 1:1); UV (MeCN/H_2_O) λmax (%) 220 (100%), 310 (33%) nm; 
(+)-HRESIMS: m/z 1233.2096 [M+H]+ (calcd for C_53_H_49_N_14_O_10_S_6_, 
1233.2080); ^1^H and ^13^C NMR data, Appendix A.

GE2270F_4_ (**4**): White amorphous powders; ${\left[\alpha \right]}_{D}^{23}$ + 88 (c 0.3, CH_2_Cl_2_:MeOH 
= 1:1); UV (MeCN/H_2_O) λmax (%) 220 (100%), 310 (33%) nm; 
(+)-HRESIMS: *m*/*z* 1277.2336 [M+H]+ (calcd for C_55_H_53_N_14_O_11_S_6_, 
1277.2337); ^1^H and ^13^C NMR data, Appendix A.

Compound **5**: White amorphous powders; ${\left[\alpha \right]}_{D}^{23}$ + 102 (c 0.3, CH_2_Cl_2_:MeOH 
= 1:1); UV (MeCN/H_2_O) λmax (%) 221 (100%), 310 (33%) nm; 
(+)-HRESIMS: *m*/*z* 1264.2495 [M+H]+ (calcd for C_54_H_54_N_15_O_10_S_6_, 
1264.2502); ^1^H and ^13^C NMR data, see Appendix A.

Compound **6**: White amorphous powders; ${\left[\alpha \right]}_{D}^{23}$ + 79 (c 0.2, CH_2_Cl_2_:MeOH 
= 1:1); UV (MeCN/H_2_O) λmax (%) 220 (100%), 310 (33%) nm; 
(+)-HRESIMS: *m*/*z* 1308.2770 [M+H]+ (calcd for C_56_H_58_N_15_O_11_S_6_, 
1308.2764); ^1^H and ^13^C NMR data, see Appendix A.

Compound **7**: White amorphous powders; ${\left[\alpha \right]}_{D}^{23}$ + 72 (c 0.2, CH_2_Cl_2_:MeOH 
= 1:1); UV (MeCN/H_2_O) λmax (%) 222 (100%), 308 (33%) nm; 
(+)-HRESIMS: *m*/*z* 1095.1655 [M+H]+ (calcd for C_47_H_43_N_12_O_8_S_6_, 
1095.1651); ^1^H and ^13^C NMR data, see Appendix A.

Compound **8**: White amorphous powders; ${\left[\alpha \right]}_{D}^{23}$ + 36 (c 0.2, CH_2_Cl_2_:MeOH 
= 1:1); UV (MeCN/H_2_O) λmax (%) 221 (100%), 307 (33%) nm; (+)-HRESIMS: *m*/*z* 1139.1898 [M+H]+ (calcd for C_49_H_47_N_12_O_9_S_6_, 
1139.1913); ^1^H and ^13^C NMR data, see Appendix A.

Compound **9**: White amorphous powders;${\left[\alpha \right]}_{D}^{23}$ + 24 (c 0.2, CH_2_Cl_2_:MeOH 
= 1:1); UV (MeCN/H_2_O) λmax (%) 221 (100%), 308 (33%) nm; 
(+)-HRESIMS: *m*/*z* 1125.1740 [M+H]+ (calcd for C_48_H_45_N_12_O_9_S_6_, 
1125.1757); ^1^H and ^13^C NMR data, see Appendix A.

### 3.6. Biological Assays

Isolated compounds of interest were tested against five microbial strains for antimicrobial testing, which were *Acinetobacter baumannii* (ATCC^®^ 19606™), *Klebsiella aerogenes* (ATCC^®^ 13048™), *Pseudomonas aeruginosa* (ATCC^®^ 9027™) *Staphylococcus aureus* Rosenbach (ATCC^®^ 25923™) and *Aspergillus fumigatus* (ATCC^®^ 46645™). The minimum inhibition concentration (MIC) and the minimum bactericidal/fungicidal concentration (MBC/MFC) were carried out using the microbroth dilution method according to the Clinical Laboratory Standards Institute (CLSI) guidelines, with some modifications. To establish the MIC values, the bacterial cells were seeded at a concentration of 5.5 × 10^5^ cells/mL and fungal spores at a concentration of 2.5 × 10^4^ spores/mL. The tested compounds were then incubated together with bacterial cells at 37 °C for 24 h and with fungal spores at 25 °C for 72 h, respectively. The OD_600_ measurements were subsequentially carried out to evaluate the inhibitory effect of the compounds. To further determine the bactericidal and fungicidal effects of the compounds, 5 µL of the treated culture was transferred onto new media microplates. The plates were incubated under the same condition, followed by the OD_600_ measurement. The cytotoxicity effects of the isolated compounds were also tested on A549 human lung carcinoma cells (ATCC^®^ CCL-185™), where cells were seeded at 3.3 × 10^4^ cells/mL. The cells were then treated with the compounds for 72 h at 37 °C in the presence of 5% CO_2_. Any Cytotoxic effect was detected with PrestoBlue™ cell viability reagent (ThermoFisher Scientific, Waltham, MA, USA). The cells were read with a fluorescence reading at an excitation of 560 nm and an emission 590 nm. Standard inhibitors, gentamicin (Gibco, Waltham, MA, USA), amphotericin (Sigma-Aldrich, St. Louis, MO, USA) and puromycin (Sigma-Aldrich, St. Louis, MO, USA) were used as the assay controls for the antibacterial, antifungal and cytotoxicity assay. All compounds were tested in triplicates to the ensure reproducibility of the results. GraphPad Prism 8 software (GraphPad, San Diego, CA, USA) was used for the analysis of the bioactivity to determine the respective IC_90_ and IC_50_ values.

### 3.7. GNPS Molecular Networking

An LC-MS/MS data file (.d) created from the Agilent QTOF mass spectrometer were converted to .mgf file formats with an Agilent Qualitative 10.0 and uploaded to the GNPS Web platform (http://gnps.ucsd.edu., accessed on 20 December 2021) for the classical molecular networking generation. MS-Cluster (0.1 Da tolerance) and a 0.02 Da tolerance for fragment ions were applied to create the consensus parent mass spectra. A network was generated where there were more than six matched fragment ions and the edges were filtered to have a minimal cosine score of 0.7. A maximum size of a molecular family was also set to 100. The output molecular networking was visualized and analyzed using Cytoscape 3.9.0. The GNPS data can be found at https://gnps.ucsd.edu/ProteoSAFe/status.jsp?task=e9f7b2ac30bd4ddabcc847f44b089a4b, accessed on 20 December 2021. 

## 4. Conclusions

*Nonomuraea jiangxiensis* strain A7611 isolated from a soil sample collected in Singapore was found to produce a class of thiopeptide GE2270, compounds **1**–**9** that were not only non-cytotoxic to laboratory human cell line A549, but also exhibited antibacterial activity against Gram-positive *S. aureus* Rosenbach, with MIC_90_ values ranging from 2 µM to 11 µM. Thiopeptides **1**–**9** shared similar structures, possessing various substitutions on thiazole rings A, D and E. Structure–activity relationship studies showed the presence of both R2 (i.e., methylene-oxy-methyl) and R3 (i.e., methyl) groups on thiazole ring D and E, respectively, which was important for their antibacterial activity. This report further confirmed the potential of thiopeptides as potent antibacterial agents.

## Figures and Tables

**Figure 1 molecules-28-00101-f001:**
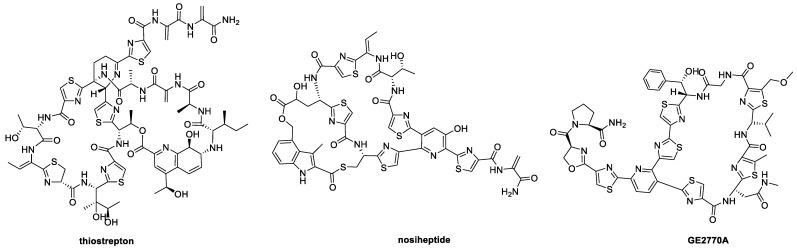
Structures of thiostrepton, nosiheptide and GE2770A.

**Figure 2 molecules-28-00101-f002:**
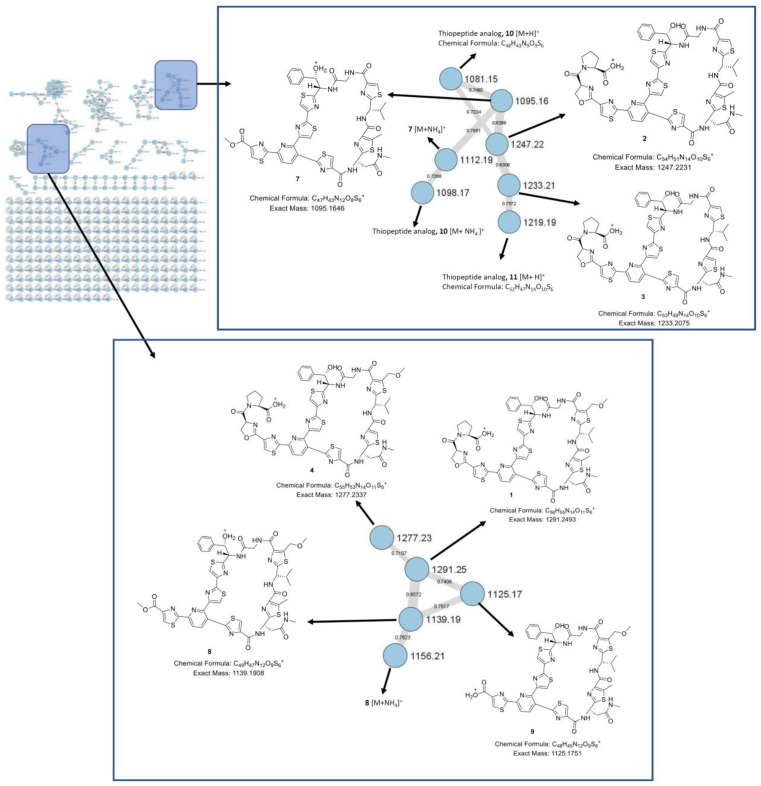
GNPS molecular networking of the crude extract of *Nonomuraea jiangxiensis* and the structures attributed to nodes in the thiopeptide GNPS clusters.

**Figure 3 molecules-28-00101-f003:**
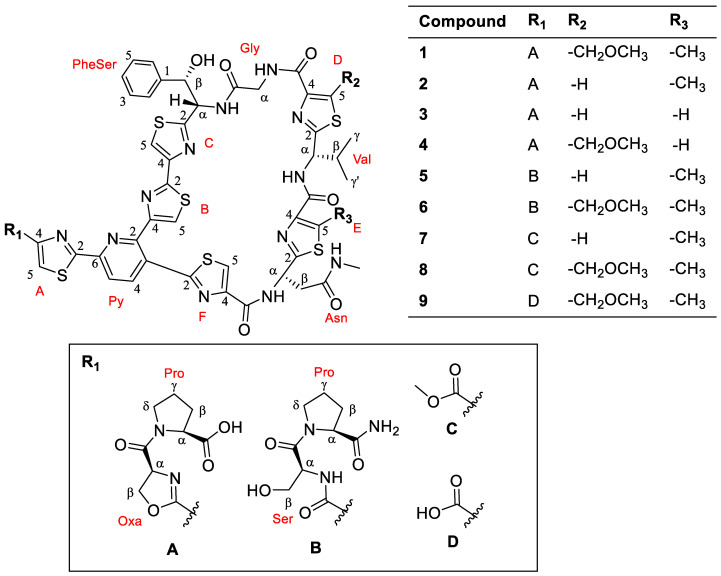
Chemical structures of compounds **1**–**9**.

**Figure 4 molecules-28-00101-f004:**
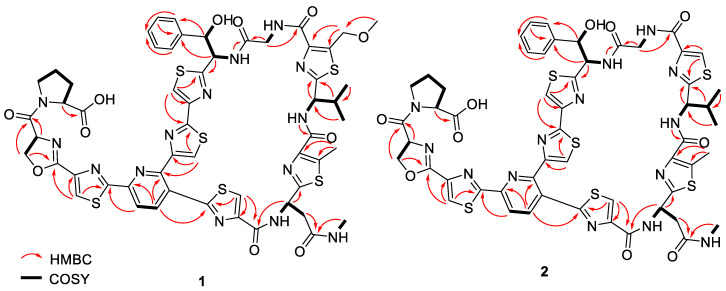
Selected COSY and HMBC correlations of **1** and **2**.

**Figure 5 molecules-28-00101-f005:**
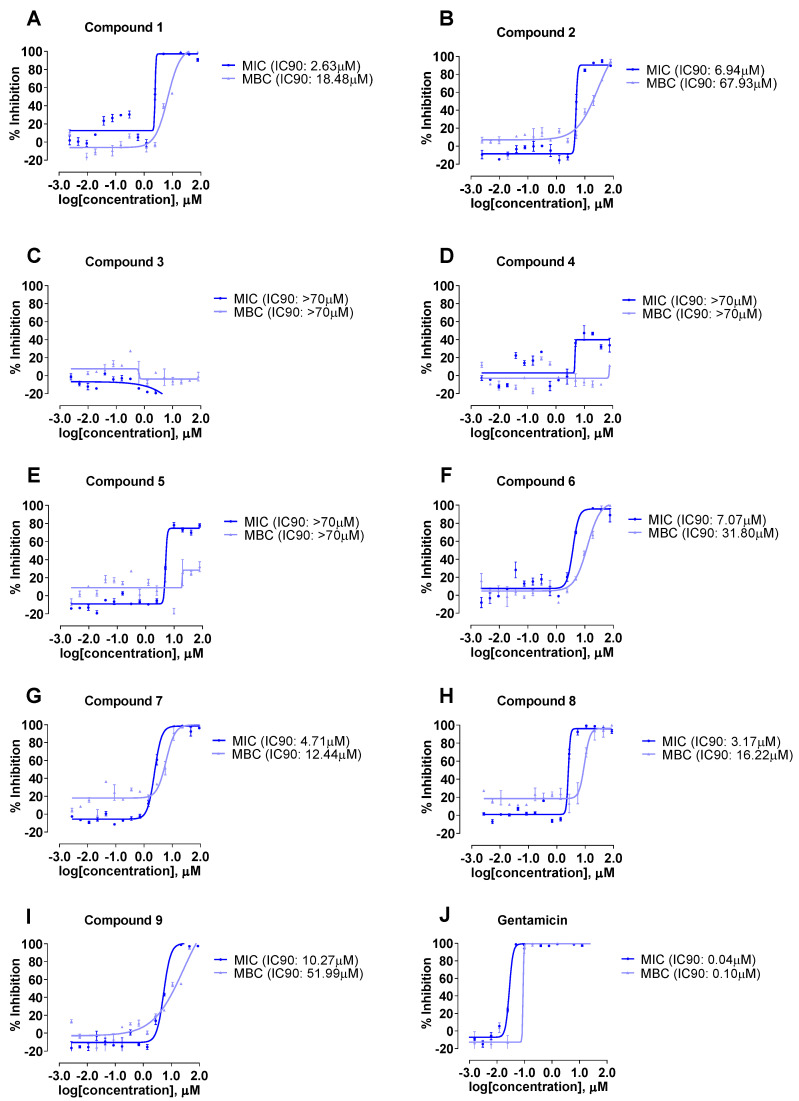
Inhibitory effect dose response curve against *Staphylococcus aureus* Rosenbach (ATCC^®^ 25923™) for compounds **1**–**9** (**A**–**I**) and positive control gentamicin (**J**).

**Figure 6 molecules-28-00101-f006:**
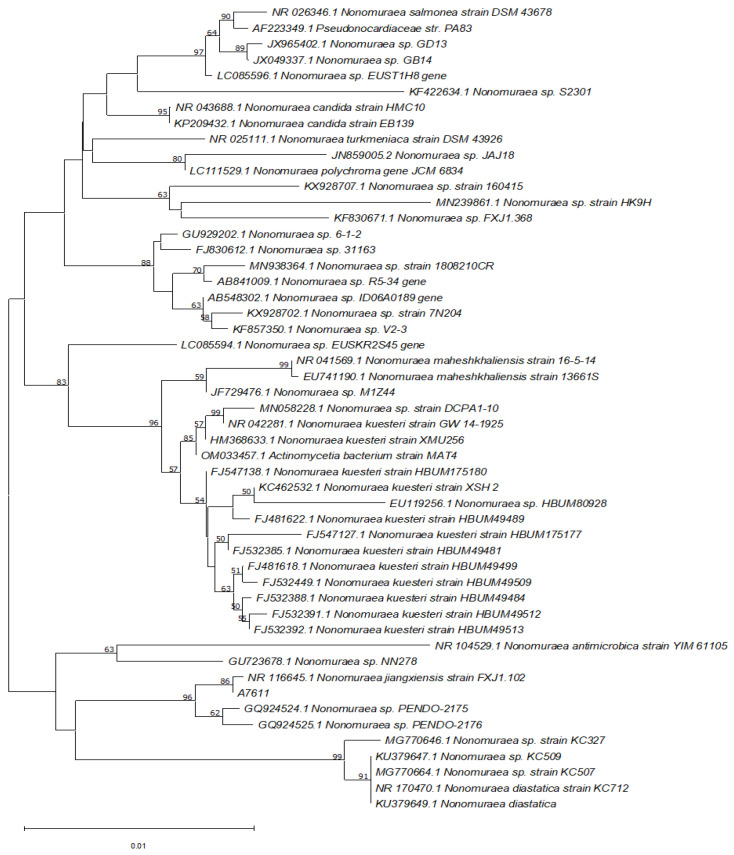
Phylogenetic tree showing the evolutionary relationship between A7611 and other type species of the genus *Nonomuraea*. Neighbor-joining phylogenetic tree was constructed based on 16S rRNA gene sequence showing the relationship between isolated strain A7611 and representatives or related actinobacteria strains retrieved from the GenBank with their respective accession numbers. Bootstrap values greater than 50% are shown at the number on the branches nodes which were analysed based on 1000 replicates. Bar, 0.01 substitutions per nucleotide position.

**Table 1 molecules-28-00101-t001:** NMR spectral data of thiopeptides **1** and **2**.

Residue	Pos.	1		2	
		^13^C	^1^H, Mult. (*J* = Hz)	^13^C	^1^H, Mult. (*J* = Hz)
Proline (Pro)	α	58.5, CH	4.25, t (8.2)	58.5, CH	4.24, m
	β	27.9, CH_2_	1.92, m, 2.19, m	27.9, CH_2_	1.91, m, 2.19, m
	γ	22.1, CH_2_	1.84, m, 1.91, m	22.1, CH_2_	1.84, m, 1.91, m
	δ	44.8, CH_2_	3.41, m, 3.50, m	44.8, CH_2_	3.39, m, 3.50, m
	CO	169.3, C	-	169.3	-
Oxazoline (Oxa)	α	53.9, CH	4.55, m	53.9, CH	4.55, m
	β	63.2, CH_2_	4.53; 4.68, dd (14.9, 6.7)	63.2, CH_2_	4.53; 4.68, dd (14.9, 6.7)
	CN	160.1, C	-	160.1, C	-
	CO	163.2, C	-	163.2, C	-
Thiazole A	2	167.9, C	-	168.0, C	-
	4	147.2, C	-	147.2, C	-
	5	132.7, CH	8.76, s	132.7, CH	8.75, s
Pyridine (Py)	2	150.1, C	-	150.1, C	-
	3	127.6, C	-	127.70, C	-
	4	141.3, CH	8.44, d (8.2)	141.3, CH	8.43, d (8.1)
	5	118.4, CH	8.27, d (8.1)	118.3, CH	8.27, d (8.1)
	6	150.0, C	-	150.0, C	-
Thiazole B	2	160.3, C	-	160.4, C	-
	4	153.1, C	-	153.1, C	-
	5	123.0, CH	8.29, s	123.0, CH	8.29, s
Thiazole C	2	170.8	-	170.8, C	-
	4	146.6	-	146.6, C	-
	5	116.2, CH	7.34, s	116.3, CH	7.34, s
Phenylserine (PheSer)	α	58.2, CH	5.24, dd (14.0, 6.8)	58.2, CH	5.24, m
	β	73.3, CH	5.03, m	73.3, CH	5.04, m
	1	141.6		141.6, C	-
	2,6	126.5, CH	7.29, m	126.5, CH	7.28, m
	3,5	127.7, CH	7.28, m	127.75, CH	7.26, m
	4	127.4, CH	7.28, m	127.4, CH	7.27, m
	NH		9.29, d (7.7)	-	9.22, d (7.9)
	OH	6.43, br s		-	6.35, br s
Glycine (Gly)	α	41.0, CH_2_	3.81, dd (17.1, 3.9); 4.32, t (17.1, 8.7)	41.3, CH_2_	3.88, dd (16.9, 4.3); 4.24, t (7.6)
	CO	169.4, C	-	169.3, C	-
	NH	-	8.45, m	-	8.55, m
Thiazole D	2	165.4, C	-	168.3,	-
	4	143.5, C	-	148.4, C	-
	5	140.8, C	-	124.5, CH	8.28, s
	5-CH_2_	67.2, C	4.99, s	-	-
	5-CH_2_O**Me**	58.4, C	3.39, s	-	-
	CO	161.2, C	-	160.3, C	-
Valine (Val)	α	55.2, CH	5.20, dd (4.7, 8.0)	55.3, CH	5.25, m
	β	33.8, CH	2.18, m	34.0, CH	2.19, m
	γ	18.3, CH_3_	0.86, d (6.9)	18.4, CH_3_	0.86, d (6.8)
	γ’	17.8, CH_3_	0.89, d (6.9)	17.8, CH_3_	0.89, d (6.8)
	NH	-	8.69, d (7.7)	-	8.75, m
Thiazole E	2	168.3, C	-	167.9, C	-
	4	141.9, C	-	141.7, C	-
	5	139.3, C	-	139.5, C	-
	5-Me	11.8, CH_3_	2.59, s	11.8, CH_3_	2.60, s
	CO	161.0, C	-	161.0, C	-
Asparagine (Asn)	α	47.9, CH	5.29, dt (4.1, 8.7)	47.9, CH	5.35, m
	β	37.4, CH_2_	1.31, m, 2.74, dd (4.1, 16.8)	37.6, CH_2_	1.47, m, 2.74, dd (3.8, 17.1)
	**CO**NHMe	169.6, C	-	169.6, C	-
	**NH**Me	-	7.41, q (4.5)	-	7.48, m
	Me	25.6, CH_3_	2.47, d (4.5)	25.6, CH_3_	2.46, d (4.5)
	NH	-	8.69, d (7.7)	-	8.74, m
Thiazole F	2	164.5, C	-	164.6, C	-
	4	149.2, C	-	149.3, C	-
	5	126.8, CH	8.61, s	126.8, CH	8.60, s
	CO	160.2, C	-	160.2, C	-

^1^H (400 MHz) and ^13^C (100 MHz) in DMSO-*d_6_*. Assignments based on COSY, HSQC and HMBC and a comparison with the literature compounds. Chemical shifts (δ) in ppm. s: singlet; br s: broad singlet; d: doublet; br d: broad doublet; t: triplet, m: multiplet.

**Table 2 molecules-28-00101-t002:** Summary of thiopeptides identified using molecular networking and manual observations of the data.

Thiopeptide	Measured Mass [M+H]^+^	Theoretical Mass [M+H]^+^	Δppm	Formula [+H]^+^
**1**	1291.2544	1291.2493	3.91	C_56_H_55_N_14_O_11_S_6_
**2**	1247.2240	1247.2331	0.69	C_54_H_51_N_14_O_10_S_6_
**3**	1233.2096	1233.2075	1.72	C_53_H_49_N_14_O_10_S_6_
**4**	1277.2336	1277.2337	−0.08	C_55_H_53_N_14_O_11_S_6_
**5**	1264.2495 *	1264.2497	−0.15	C_54_H_53_N_15_O_10_S_6_
**6**	1308.2770 *	1308.2759	0.84	C_56_H_58_N_15_O_11_S_6_
**7**	1095.1655	1095.1646	0.86	C_47_H_43_N_12_O_8_S_6_
**8**	1139.1898	1139.1908	−0.85	C_49_H_43_N_12_O_8_S_6_
**9**	1125.1740	1125.1751	−1.00	C_48_H_45_N_12_O_9_S_6_
**10**	1081.1486 ^#^	1081.1502	−1.53	C_48_H_43_N_9_O_9_S_6_
**11**	1219.1926 ^#^	1219.1918	0.63	C_52_H_47_N_14_O_10_S_6_
GE2270A	1290.2663	1290.2653	0.75	C_56_H_56_N_15_O_10_S_6_

* Unconnected singleton nodes in molecular network, ^#^ Uncertain structures.

**Table 3 molecules-28-00101-t003:** Biological activities of compounds **1**–**9** and positive control gentamicin against *Staphylococcus aureus* Rosenbach (ATCC^®^ 25923™).

Thiopeptide Compounds	MIC_90_ [(μM) mean ± SD]	MBC_90_ [(μM) mean ± SD]
**1**	2.63 ± 0.12	18.48 ± 1.34
**2**	6.94 ± 0.83	67.93 ± 1.55
**3**	–	–
**4**	–	–
**5**	–	–
**6**	7.07 ± 0.55	31.80 ± 1.38
**7**	4.71 ± 0.37	12.44 ± 1.06
**8**	3.17 ± 0.28	16.22 ± 1.48
**9**	10.27 ± 0.75	51.99 ± 1.41
Gentamicin	0.04 ± 0.01	0.10 ± 0.01

Values are expressed as mean ± SD in triplicates. (–) Compounds show no inhibition for MIC_90_ and MBC_90_ at 70 µM concentration.

## Data Availability

Data is contained within the article or Appendix A.

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
