# Peer review of "Antibacterial Thiopeptide GE2270-Congeners from Nonomuraea jiangxiensis"

_molecules, 2022, doi:10.3390/molecules28010101_

Round 1

Reviewer 1 Report

The manuscript entitled ''Antibacterial Thiopeptide GE2270-Congeners from Nonomuraea jiangxiensis'' by Kuan-Chieh Ching et al is piece of research work but definitely in the current form, it lacks novelty. 

In general, authors had put lots of effort to carry out the experiments involved in this research work, but the presentation is too poor.

As a reader, the current form of the manuscript does not clearly reveal the exact aim of the study. For example, there was no detail information given on the bacterial strain used in this study. In the material and method section, it's all started with growing the strain on Bennett's agar without revealing any background details of the sample or isolation procedure. The way authors have presented the data, that clearly tells, neither the isolated compounds were coming through genome mining approach, nor they came through bioactivity guided isolation process. In that case, thiopeptides might have been isolated before the bacterial strain was characterized. So, the writing should have followed the same pattern i.e., isolation, characterization of thiopeptides should be mentioned before and followed by characterization of bacteria. On the other hand, as author mentioned, some of these compounds were previously chemically synthesized or isolated from engineered strains, in this attempt compounds were isolated from native producer, but no such new extraction/isolation/characterization methods has been used to isolate them which lacks novelty. No positive controls were used for biological activity studies against pathogens, that again not a good way of presenting data. So only based on the fact that compounds were isolated from its native producer, which was not reported before, this manuscript cannot be accepted in Molecules. 

Author Response

The manuscript entitled ''Antibacterial Thiopeptide GE2270-Congeners from Nonomuraea jiangxiensis'' by Kuan-Chieh Ching et al is piece of research work but definitely in the current form, it lacks novelty.

In general, authors had put lots of effort to carry out the experiments involved in this research work, but the presentation is too poor.

As a reader, the current form of the manuscript does not clearly reveal the exact aim of the study.

  • We thank the reviewer for the comments. We believed we have stated the aim clearly in our initial submission “As part of our on-going studies in biologically active secondary metabolites from Actinobacteria, several thousand extracts from our in-house library were screened for their antibacterial activity and a few extracts demonstrated antibacterial activity against Staphylococcus aureus.”

For example, there was no detail information given on the bacterial strain used in this study.

  • We thank the reviewer for the comments. The detailed information of the bacterial strain used in the study was also stated in the manuscript as shown by this statement “A nucleotide BLAST search of the 16S rRNA gene sequence of A7611 was performed against the NCBI 16S ribosomal RNA database revealed that the isolate shared 99.93% sequence identity (E-value = 0.0) to 16S rRNA of Nonomuraea jiangxiensis with accession number NR_116645.1”. This strain belongs to genus Nonomuraea and species

In the material and method section, it's all started with growing the strain on Bennett's agar without revealing any background details of the sample or isolation procedure. The way authors have presented the data, that clearly tells, neither the isolated compounds were coming through genome mining approach, nor they came through bioactivity guided isolation process. In that case, thiopeptides might have been isolated before the bacterial strain was characterized.

  • Thank you for the insightful comments. We have included the origin and background on this strain in section 2.2. The bacterial strain A7611 was isolated from terrestrial soil in Singapore. As explained in Ln 40-48, A7611 is one of the strains whose extracts were identified as possessing antibacterial activity through our on-going bioactivity screening study. This strain was selected for further study after HR-ESIM, Global Natural Products Social (GNPS) molecular networking and chemical analysis on these antibacterial extracts. Hence, the isolation of these thiopeptides resulted from a combination of bioactivity assessment and chemical analysis.

So, the writing should have followed the same pattern i.e., isolation, characterization of thiopeptides should be mentioned before and followed by characterization of bacteria.

  • Very good suggestion. We have amended the flow of the manuscript following the reviewer’s suggestion. The strain characterization discussion has been moved to the end of the manuscript.

On the other hand, as author mentioned, some of these compounds were previously chemically synthesized or isolated from engineered strains, in this attempt compounds were isolated from native producer, but no such new extraction/isolation/characterization methods has been used to isolate them which lacks novelty.

  • We thank the reviewer for the comments. Even though several compounds (compounds 1, 6, 8, and 9) in this study have been previously reported from engineered strains, we also discovered previously undescribed compounds (compounds 2, 3, 4, 5, and 7). The novelty of this study lies in the discovery of new thiopeptides, which not only enriched the structural diversity of natural thiopeptides but also provided insights towards structure activity relationships. Moreover, this is the first report of isolation of compounds 1, 6, 8 and 9 from a native strain, which we believe is a novelty itself. Thus, we disagree with the reviewer on their comment about the lack of novelty.

No positive controls were used for biological activity studies against pathogens, that again not a good way of presenting data.

  • We thank the reviewer for the comments. We have stated in section 2.6: Biological assays in our original submission that we used gentamicin, amphotericin, and puromycin as the positive controls for the antibacterial, antifungal and cytotoxicity assays. To further address the reviewer’s comment, we have added the bioactivity data of the positive control, gentamicin in the manuscript as shown in Table 3 and Figure 5 in the results and discussion.

Reviewer 2 Report

The paper introduces a meaningful work, which is sugested for publication after minor revision:

1. What about the extraction of the dried samples by MeOH? and the yield (%) of MeOH extract together with its weight?

2. What was the detection wavelength used in preparative HPLC?

3. Please provide the retention time of compound 1,2,7,8,9.

4. The difference between the calculated and obtained m/z for compound 1,3,8 & 9 in page 4 or table 2 is a little great.

5. How about the purities of standard inhibitors?

6. The measurement of melting point range is suggested for all the compounds.

7. α-H in Oxazoline should be dd peaks, please check. The β-protons are two non-identical protons. α-H in Asparagine also needs confirmation.

8. Fig.6 is blurred.

Author Response

Reviewer 2:

The paper introduces a meaningful work, which is suggested for publication after minor revision:

  1. What about the extraction of the dried samples by MeOH? and the yield (%) of MeOH extract together with its weight?

We thank the reviewer for the question. The crude extract obtained from 2L fermentation consists of broth extract 84.4% (13.25 g) and biomass extract 15.6% (2.45 g). Combined weight of these extracts is 15.70 g. We have added this information in section 2.3: Fermentation and extraction of bacterial crude extract.

  1. What was the detection wavelength used in preparative HPLC?

The detection wavelength used in the preparative HPLC was 254 nm. We have added this statement in section 2.1: General experimental procedures.

  1. Please provide the retention time of compound 1,2,7,8,9.

We have provided the retention time (RT) of compounds 1, 2, 7, 8 and 9 in section 2.4: Isolation and structure elucidation.

RT of compound 1 = 37 min

RT of compound 2 = 30.5 min

RT of compound 7 = 45 min

RT of compound 8 = 51 min

RT of compound 9 = 39 min

  1. The difference between the calculated and obtained m/z for compound 1,3,8 & 9 in page 4 or table 2 is a little great.

We thank the reviewer for the comments. The instructions for authors in the MDPI molecules website stated that physical and spectroscopic data should be prepared according to ACS's Preparation and Submission of Manuscripts standard, and it was stated in the ACS author’s guidelines that “the structures of compounds are expected to be supported by high-resolution mass spectrometry (error limit 5 ppm or 0.003 m/z units)”.

The error limits showed in Table 2 is within the acceptable error limit of 5 ppm stated by ACS author’s guidelines. In addition, we have calculated the error limits between the calculated and obtained m/z for compounds 1, 3, 8 and 9. All of them are within the acceptable error limit of 5 ppm.

Compound 1 = 3.48 ppm

Compound 3 = 1.29 ppm

Compound 8 = -1.32 ppm

Compound 9 = -1.51 ppm

  1. How about the purities of standard inhibitors?

We thank the reviewer for the questions. The purities of the standard inhibitors are gentamicin: ~90% (HPLC), amphotericin: ~80% (HPLC) and puromycin:  ≥ 98% (HPLC). These are purchased standard inhibitors from vendors.

  1. The measurement of melting point range is suggested for all the compounds.

We thank the reviewer for the comments. We have checked the instructions for authors in the MDPI molecules website and it stated that physical and spectroscopic data should be prepared according to ACS's Preparation and Submission of Manuscripts standard (page 4).

It was stated in the ACS author’s guidelines that “melting point determinations should not be provided for compounds described as amorphous solids.” Therefore, we did not provide the melting points for these new compounds because they were isolated as amorphous powders.

  1. α-H in Oxazoline should be dd peaks, please check. The β-protons are two non-identical protons. α-H in Asparagine also needs confirmation.

We thank the reviewer for pointing this out. We agree with the reviewer that the β-protons in oxazoline should be non-identical as the bond between C-α and C-β is not rotatable due to the constrained ring system. Unfortunately, attempts to determine the two coupling constants that corresponds to a dd peak for α-H proved unsuccessful due to the poor resolution of the signal. Hence, we amended the coupling pattern as a multiplet (m) instead.

Due to the rotatable nature of bond between C-α and C-β in Asparagine, the β-protons are two identical protons, hence a dt coupling pattern was observed as α-H in Asparagine couple with two β-protons and one N-H proton.

  1. Fig.6 is blurred.

We thank the reviewer for the comment. Graph is edited to 2 columns where some redundant labelling has been removed in the figure and included in the figure captions (as shown below). Amended subfigures are presented as Figure 5 in the manuscript.

Reviewer 3 Report

The manuscript “Antibacterial Thiopeptide GE2270-Congeners from Nonomuraea jiangxiensis" is devoted to the isolation and characterization of various thiopeptides from the bacterial strain. Nine new naturally occurring thiopeptides were isolated and characterized by NMR and LC-MS. Antibacterial activity against to five bacterial strains were tested. Obtained data may be useful for potent antibacterial agents development.

The manuscript is written in good style, the results are described in detail, and exhaustive conclusions are given.

I think, this manuscript can be published in the Molecules after minor revision

  1. Methods of statistical data processing should be presented, SD values should be added.
  2. Figure 6. is unreadable. I recommend changing them.

Author Response

Reviewer 3:

The manuscript “Antibacterial Thiopeptide GE2270-Congeners from Nonomuraea jiangxiensis" is devoted to the isolation and characterization of various thiopeptides from the bacterial strain. Nine new naturally occurring thiopeptides were isolated and characterized by NMR and LC-MS. Antibacterial activity against to five bacterial strains were tested. Obtained data may be useful for potent antibacterial agents development.

The manuscript is written in good style, the results are described in detail, and exhaustive conclusions are given.

I think, this manuscript can be published in the Molecules after minor revision.

  1. Methods of statistical data processing should be presented, SD values should be added.

We thank the reviewer for the comment and with that we have edited the table of interest (Table 3) to include the standard deviation of respective IC90 results of each compound and positive control.

Table 3. Biological activities of compounds 19 and positive control gentamicin against Staphylococcus aureus Rosenbach (ATCC® 25923™)

Thiopeptide Compounds

MIC90

[(μM) mean ± SD]

MBC­90

[(μM) mean ± SD]

1

2.63 ± 0.12

18.48 ± 1.34

2

6.94 ± 0.83

67.93 ± 1.55

3

4

5

6

7.07 ± 0.55

31.80 ± 1.38

7

4.71 ± 0.37

12.44 ± 1.06

8

3.17 ± 0.28

16.22 ± 1.48

9

10.27 ± 0.75

51.99 ± 1.41

Gentamicin

0.04 ± 0.01

0.10 ± 0.01

Values are expressed as mean ± SD in triplicates.

(–) Compounds show no inhibition for MIC90 and MBC90 at 70 µM concentration.

  1. Figure 6. is unreadable. I recommend changing them.

We thank the reviewer for the comment. Graph is edited to 2 columns where some redundant labelling has been removed in the figure and included in the figure captions. Amended subfigures are presented as Figure 5 in the manuscript.

Figure 5. Inhibitory effect dose response curve against Staphylococcus aureus Rosenbach (ATCC® 25923™) for compounds 1–9 and positive control gentamicin (Figures A–J). 

Round 2

Reviewer 1 Report

I am happy with the justification given by the authors in the revised version of this manuscript and also satisfied with corrections made by the authors. Therefore, I feel the manuscript is good to go for the publication in its present form.